# Therapeutic relationships within child and adolescent mental health inpatient services: A qualitative exploration of the experiences of young people, family members and nursing staff

Samantha Hartley[1,2]*, Tomos Redmond[1], Katherine Berry[2]

**1** Pennine Care NHS Foundation Trust, Ashton-under-Lyne, United Kingdom, **2** Division of Psychology and Mental Health, School of Health Sciences, University of Manchester, Manchester, United Kingdom

* samantha.hartley@manchester.ac.uk

## Abstract

Child and adolescent mental health services (CAMHS), especially inpatient units, have arguably never been more in demand and yet more in need of reform. Progress in psychotherapy and more broadly in mental health care is strongly predicted by the therapeutic relationship between professional and service user. This link is particularly pertinent in child and adolescent mental health inpatient services where relationships are especially complex and difficult to develop and maintain. This article describes a qualitative exploration of the lived experienced of 24 participants (8 young people, 8 family members/carers and 8 nursing staff) within inpatient CAMHS across four sites in the UK. We interviewed participants individually and analysed the transcripts using thematic analysis within a critical realist framework. We synthesised data across groups and present six themes, encapsulating the intricacies and impact of therapeutic relationships; their development and maintenance: *Therapeutic relationships are the treatment*, *Cultivating connection*, *Knowledge is power*, *Being human*, *The dance*, and *It's tough for all of us in here*. We hope these findings can be used to improve quality of care by providing a blueprint for policy, training, systemic structures and staff support.

## Introduction

The mental health needs of young people are growing steadily and a cause for concern [1, 2]. Despite commitment to improved access to community services [3], some young people continue to require support by specialist inpatient mental health facilities, with evidence of alternative provision still limited [4]. Internationally, inpatient care faces huge challenges alongside growing demand [5]. Particularly in the UK, where this paper is focused, there is a drive for quality improvement. The UK Future in Mind report [6] declared that difficulties in appropriate access persist and the pressure of escalating referrals, alongside increased complexity, is

or first author, in line with individual consent given by participants (see response to reviewers document). The study sponsor is Pennine Care NHS Foundation Trust, Simon Kaye, Research and Innovation Manager, email: researchdevelopment. penninecare@nhs.net.

**Funding:** Dr Hartley is funded by a National Institute for Health Research (NIHR) Integrated Clinical Academic Clinical Lectureship for this research project. Grant number: ICA-CL-2017-03-008. www.nihr.ac.uk. This paper presents independent research funded by the National Institute for Health Research (NIHR). The views expressed are those of the author(s) and not necessarily those of the NHS, the NIHR or the Department of Health and Social Care. The funder played no other role.

**Competing interests:** The authors have declared that no competing interests exist.

contributing to a rise in lengths of stay. Inpatient care can lead to improvements in health across a range of need areas [7], although the nature of the service provided is crucial [8]. The Royal College of Psychiatrists have called for improved access to high quality inpatient services for children and young people and the Department of Health have prioritised the development of dedicated inpatient services for 16–25 year olds [3]. In 2019, NHS England established a national quality improvement taskforce dedicated to inpatient mental health care, such is the level of need and concern [9]. However, questions remain: what is 'high quality' and what do young people need to support their care in these environments?.

Therapeutic relationships, or the alliance built between professional care giver and care receiver, is the strongest predictor of good treatment outcomes, no matter what intervention model or approach is utilised [10, 11]. A recent meta-analytic review demonstrated that the alliance is positively related to outcome above and beyond patient characteristics and allied treatment processes such as adherence or competency [12]. Particularly in child and adolescent mental health services (CAMHS), the alliance between the child or family and staff is the key predictor of positive outcomes [13, 14], yet is often a factor neglected by research endeavours [15]. Working systemically and in partnership with parents and carers is essential, although often complicated by the family's own history with services and relational dynamics [16].

The National Institute for Health Research recently identified the quality of relationships as one of four critical influences on young people's experience of inpatient mental health care [17]. Specifically, young people identify that relationships with staff can ease the experience of admission [18, 19] and they especially appreciate professionalism and expertise in the context of reciprocal engagement; making a 'human connection' [20]. The relationship between staff members and young people can contribute towards the inpatient ward's therapeutic milieu that aids recovery even when young people struggle to access direct interventions, i.e. one-to-one therapy [21]. Positive therapeutic alliance- alongside lengths of stay and preadmission functioning- independently predicts outcomes of admissions [22]. Despite its importance, poor alliance is the main factor for ending therapeutic treatment [23], with various challenges to its development.

When supporting young people, there is a need to navigate issues of power and control [24] and manage the complexities, emotional and relational experiences or 'push and pull' of working with adolescents who experience disrupted ways of relating to people due to their early relationships [25]. The alliance is difficult for young people who have been maltreated [26] and the developmental trend toward increasing autonomy from adults can represent an additional obstacle [27]. Health professionals have a key role to play in the development of alliance [28] and young people, carers and staff can agree on the necessary aspects of its definition, creation and maintenance [29]. Adolescence is a time of rapid and multifaceted change [30], accompanied by relational challenges. When coupled with additional trauma (historical or iatrogenic) and mental health needs, it is not surprising that the process of developing therapeutic relationships in this context is tricky for all involved.

Within inpatient settings, workforce issues have also been highlighted, with services seeing severe problems with both recruitment and retention; arguably influenced by staff working in the context of rising levels of harm, violence and restrictive interventions [31, 32]. In this context, staff members suffer high levels of burnout [33] and there are difficulties maintaining a skilled and sufficient workforce [34]. It is therefore essential that front-line professionals are well supported in managing therapeutic relationships [35], in ways based on a thorough understanding of how they are best developed and maintained. Nursing staff in particular are well-placed to build therapeutic alliance, which is core to the profession [31, 35–39]. However, evidence-based methods for supporting this work are lacking [40].

Despite the need for high quality mental health inpatient services for young people, the clear role that therapeutic relationships have in this and the apparent difficulty in developing and maintaining these, there has been no in-depth qualitative study of the alliance in this context. The current study therefore aimed to comprehensively explore the experience of therapeutic relationships between young people, their carers and nursing staff admitted to or working on adolescent inpatient mental health wards. This is the first study to qualitatively investigate the alliance in this setting from the multiple perspectives involved in therapeutic relationships here. The qualitative approach enables the synthesis of a rich dataset that permits a more in-depth understanding and thus might provide material to aid the development of interventions or structures to support future service improvements. The study proceeded in line with a critical realist perspective, which allows the influence of the specific context of inpatient units to be recognised.

## Materials and methods

### Design

The idea for this project was rooted in clinical experience and patient and public involvement in the setting of interest, utilising consultation groups with relevant stakeholders and close liaison with a young person and carers' participation council. Qualitative methodology involving semi-structured interviews was used to elicit the experiences of young people who have been admitted to adolescent inpatient wards, their carers or family members and nursing staff. The study is reported in line with the Consolidated criteria for Reporting Qualitative research (COREQ) Checklist [41].

### Sample

We recruited eight young people (YP), eight carers and eight nursing staff. Sample sizes were based on previous studies that have sought and thematically analysed the views of similar multiple groups [42] and in line with methodological guidance [43–45]. A sampling frame was used in order to recruit a maximum variation sample. Participants were recruited via the UK NHS and ethical approval was granted (West of Scotland REC 4; IRAS ID 246547). Young people were eligible if they were 13–18 years of age; currently admitted to an adolescent inpatient mental health facility; able to provide (or parents provide) informed consent/assent for interviews and audio recording; with sufficient English language proficiency to take part in qualitative interviews. Carers were aged 18 or over; a named carer or family member of a young person who was currently admitted to an adolescent inpatient mental health facility; with sufficient English language proficiency to take part in qualitative interviews. Staff members were aged 18 or over and a qualified staff nurse or nursing assistant working on an adolescent inpatient mental health facility. In terms of connections between participants, we neither encouraged nor prevented participants from the same family taking part, which is an approach that was approved by our ethics committee review. In terms of staffing relationships, participants were recruited across four sites and some staff will have cared for the young people interviewed. The nature of inpatient care is that all staff members will have contact/ relationships of varying degrees with all young people and family members on the unit.

### Recruitment and research procedures

The researchers liaised with clinical staff from inpatient facilities to inform them about the study. All potential participants were given a participant information sheet and the opportunity to discuss the study with the researcher, ask any questions and receive clarification. The

process for obtaining informed consent was in line with HRA Guidance and a face to face, semi-structured interview was conducted by the researcher in a private room on the hospital site. Where young people were Gillick competent, we did not seek parental consent, as per our ethical approval.

## Interview schedule and data

A flexible topic guide was used to explore participants' views about: 1) Experience of the ward; 2) Experience of relationships between young people, family members and nursing staff on the ward; 3) Barriers and facilitators of positive therapeutic relationships; 4) Impact of therapeutic relationships. Topic guides were developed from published literature and in consultation with the patient, carer and staff involvement group allied to the project, who reviewed drafts, suggested wording changes and additional areas of exploration and prompts. The topic guide was updated following the initial interviews. The interview was audio recorded on a secure dictaphone and uploaded for transcription. A study ID number was assigned to each participant and identifying information removed in the transcription stage. Transcriptions were completed and checked against the original audio by the first and second authors and volunteers listed in the acknowledgments.

## Research team and reflexivity

All interviews were conducted by the first author. Participants were made aware that the interviewer was a clinical psychologist and researcher working in an adolescent inpatient facility; the potential for this to influence the discourse (in terms of a staff member asking about relationships with staff members) was acknowledged with participants and they were reminded of confidentiality and encouraged to share as openly as they could. Some participants knew the first author in a clinical capacity and this intersection of relationship was acknowledged. On reflection, this circumstance on some occasions seemed to facilitate openness of participants (due to familiarity with the interviewer) but possibly slightly limited open exploration in questioning (for the same reason). The study proceeded in line with critical realism and contextualism perspectives. A critical realist epistemological stance was taken as this is most consistent with the aims and context of the study. This stance assumes that psychological phenomena do have some external basis in reality outside of any single individual's interpretation, but these phenomena are blurred and inherently bounded by culture and context; arguably especially important in settings such as inpatient wards and thus requiring careful consideration [46]. This framework allows broad inferences to be drawn whilst recognising the particular context of the participants [47].

## Data analysis

The interviews were recorded, transcribed, coded and analysed using thematic analysis [43–45] utilising paper, Word and Excel. Considering the limited theory in the field, the development of themes was inductive in nature, without reference to existing theories and semantic, manifest meaning was prioritised. Although it is recognised the boundary with more latent meaning is not always clear. The first author led the analysis, alongside the second, with review from the third. The analysis proceeded in the following manner: i) authors 1 and 2 familiarised themselves with one transcript (randomly selected); ii) authors 1 and 2 both independently coded this transcript line by line then met to review and discuss; iii) On the basis of this discussion, it was decided that only data directly relevant to the research question be coded; iv) author 2 then independently coded a second randomly selected transcript and reviewed this with author 1, from which initial codes were identified; v) author 2 then coded the remaining

data, utilising, adding to and adapting the initial codes; vi) authors 1 and 2 met regularly to review this coding and the data extracted; vii) Using printed paper versions of the codes, authors 1 and 2 reviewed and grouped related codes together to explore linkages and overlap in order to produce themes and subthemes; viii) authors 1 and 2 then reviewed these in relation to the original codes, making any necessary amendments, ix) author 2 then loosely filtered the quotes for inclusion within the subthemes, omitting data that was not relevant to therapeutic relationships between nursing staff and young people/family members; x) author 1 filtered this further, liberally selecting the most relevant quotes for each subtheme; xi) author 2 reviewed these selected quotes for homogeneity with the subtheme and theme definitions.

### Participant groupings

An open and curious approach was taken to the possibility of commonalities and divergences within and between the participant groupings of young people, carers and staff, and therefore whether the thematic structure would be siloed within these or cutting across. Individual codes were present across group structures and as these were developed into sub-themes and themes, the thread of shared understanding, incorporating specific experiences, was still evident. All participant groups contributed data to every code, sub-theme and theme. Therefore the results are presented with quotes and synthesis inter-weaved and with points of conflict and agreement highlighted. We feel this structure reflects the essence of the content of the data; that therapeutic relationships are developed by, and embody, a complex yet fruitful interplay between the different parties involved.

## Results

An overview of participant demographics is presented in Table 1, showing that the data represents the views of a diversity of genders, ethnicities and age groups. We were also able to recruit both young people admitted informally and those on a section, alongside staff who were qualified nurses and those who were non-registered nursing assistants. Our carer group included both mothers and step-fathers. The analysis process resulted in the development of six key themes: *Therapeutic relationships are the treatment*, *Cultivating connection*, *Knowledge is power*, *Being human*, *The dance*, and *It's tough for all of us in here*. Fig 1 outlines the sub-theme structure and Fig 2 depicts the conceptual relationship between themes. The results section is structured at the thematic level, with detail from the sub-themes woven into the analytic narrative. Quotes are provided alongside ID number and participant grouping.

**Table 1. Participant demographic data.**

|  | Carer (n = 8) | Staff member (n = 8) | Young person (YP; n = 8) |
|---|---|---|---|
| **Gender** | Female (n = 6) | Female (n = 6) | Female (n = 5) |
|  | Male (n = 2) | Male (n = 2) | Male (n = 3) |
| **Mean age** | 48 (SD 7.59, range 36–56) | 39 (SD 9.28, range 26–55) | 15.62 (SD: 1.31, range 13–17) |
| **Ethnicity (self-described)** | White British (n = 7) British Pakistani (n = 1) | White British (n = 6) Black African (n = 1) Mixed Race (n = 1) | White British (n = 7) British Pakistani (n = 1) |
| **Role** | Mother (n = 6) | Nurse (n = 3) | - |
|  | Step-father (n = 2) | Nursing Assistant (n = 5) |  |
| **Mean years working in inpatient setting** | - | 7 (SD 6.36) | - |
| **Mental Health Act status** | - | - | On section (n = 2) |
|  |  |  | Informal (n = 6) |

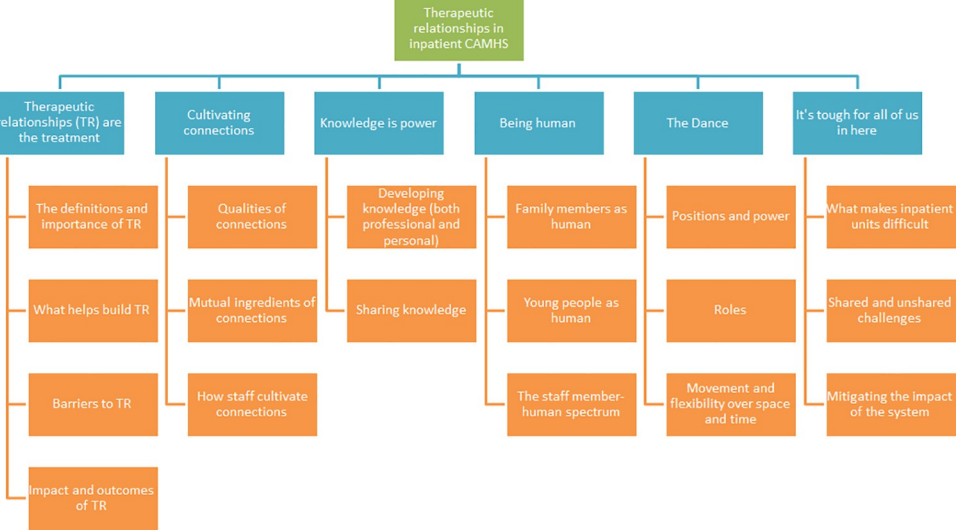

**Fig 1. Theme and sub-theme structure.**

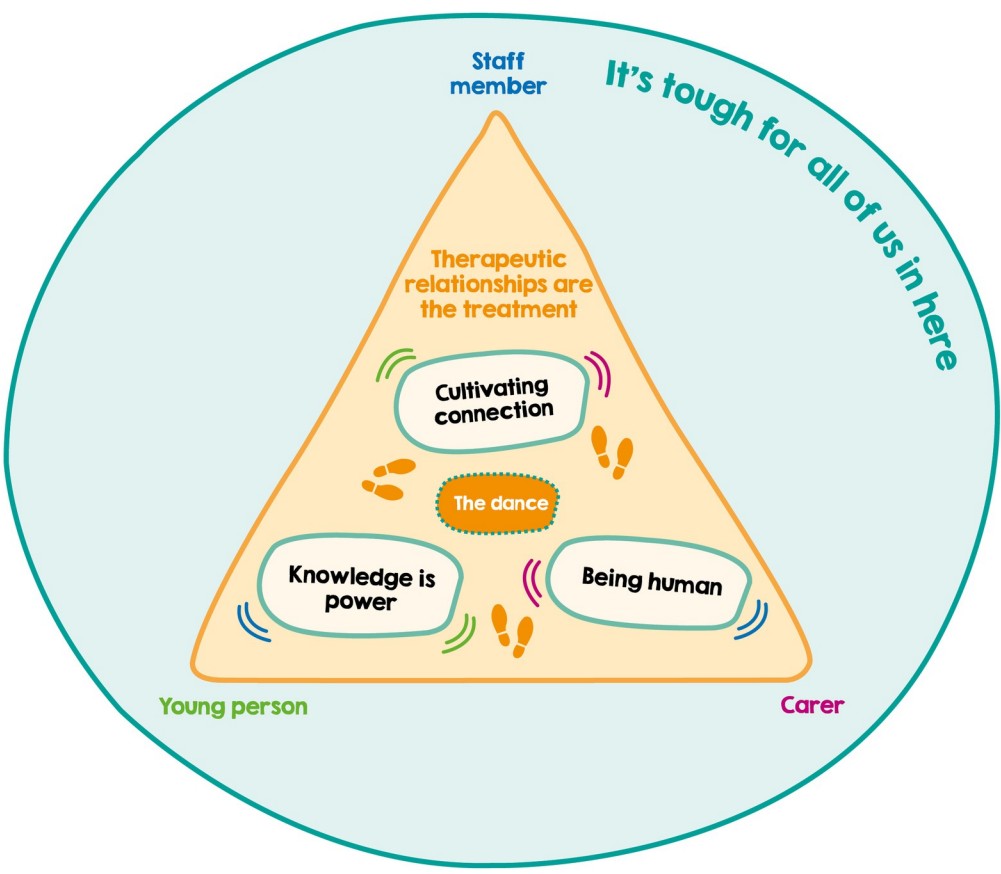

**Fig 2. Conceptual diagram of themes.**

**Theme one—Therapeutic Relationships are the Treatment:** '*cos at the end of the day it's the relationship between the patient and nurse that really matters' (20, YP).*

Therapeutic relationships are the main aspect of inpatient mental health care for young people and their families; they are not just the facilitators of other forms of care, they constitute the essence of the therapeutic milieu. However, they are not fully equivalent to other treatment; they are collaboratively built (not 'done to' or 'given to'), they require skilful development and maintenance, face unique barriers, and can have far-reaching impact on mental health and life experiences:

> *It's finding that common ground . . . how do we get through this difficult process together and how do we work together to get you [YP] out of this tricky situation, in the best way (15, Staff)*

The function of therapeutic relationships is unique and commensurate with the needs of young people requiring hospital admission; '*people that are here usually really struggle with relationships and I don't think they know what a healthy relationship is' (19, Staff).* All groups recognised the value and potential function of relationships and their variety and both their fragility and durability is also apparent, '*I feel like a lot of the times if you've built that connection it's not really gonna go . . . it sort of creates a sense of normality' (12, YP).*

Therapeutic relationships have to be maintained. They require personal qualities, understanding, team work and emotional support to sustain them, because '*starting something is different from managing' (9, Staff).* The investment of time is key, conferring value and recognising worth, otherwise the opportunity can be missed:

> *People saying if I've got time I'll speak, we'll talk later, so that's good because it's being honest about how busy everything is, but maybe if it then doesn't happen a few times I suppose the young person could get quite disappointed. (16, Carer)*

When staff members are available, this confers value and care, it '*sends a message that you care enough to follow through, and if you're consistent, after a while I think they're gonna trust you, and that's when they'll start to talk to you a bit more, and you start to get the relationship' (19, Staff).* This highlights the importance of reliability and consistency in YP feeling valued and cared for; essential for a trusting relationship. Participants felt managing the development of relationships cannot be based on a manual or protocol; '*I feel like if you're not prepared to treat people on an individual basis, based on kind of who they are, and sort of have a blanket rule, it will not work' (1, YP).* Staff elaborated this, suggesting it requires skilful use of self; responsive and flexible interactions that are dynamic and unique to each pairing and patience, '*you don't know what they have gone through so don't force relationship' (9, Staff).* Despite threats to this relationship, staff persist with developing it:

> *No matter what they do, it's not going to change how I am with them, so even though I might have had to restrain them minutes ago, the relationship hasn't changed, and I make sure they know that, so I'll go and say like, you alright, you know, then move past it. (19, Staff)*

Even once initial trust built, the work is not done; therapeutic relationships face ruptures; there are times when young people might push relationships away and '*put a wall up against everybody' (2, Staff).* Staff find themselves stuck with how to manage this while protecting the relationship–balancing the need to persist yet not intrude and eventually findings themselves exhausted and frustrated, which in turn can affect relationships and outcomes:

*sighs* *I'm tired now . . . you know this isn't working. It's repetitive like you'd half see a change like you're not seeing any changes being made, it's just the same old stuff and then the young person gets bored of it, they've heard it all before.(3, Staff)*

Judgements or perceived judgements can also pose a barrier to engagement, as young people fear that their behaviour might elicit bad feeling from staff, which in turn can damage trust. All groups recognised that this could happen after a single incident or moment; signifying fragility of the alliance and the need for dynamic, responsive and long term work in order to maintain relationships. Despite such relational challenges, participants felt there is always the potential to establish a therapeutic alliance, it just requires time and active effort by staff to foster a personal connection.

All participant groups suggested therapeutic relationships have a positive impact; *'probably one of the best things about being in here' (12, YP)*, improving mental health and positive outlook, '*like it's getting the constant reminders of them telling me we know you don't hold the hope at the moment but you can, we'll hold it for you' (6, YP)*. In time, this combats the hopelessness that often accompanies these circumstances; it *'gives us hope that they're gonna turn in the right direction and they're thinking differently, thinking more clearly and not wanting to hurt themselves' (24, Carer)*.

The alliance with staff is more than a task, it is a connection that can facilitate young people to be the best version of themselves; '*To then becoming more of the person that they want to be and sort of working things out themselves, moving forward and focusing on things and then deciding their own interests' (2, Staff)*. When the relationships do not work, it is not just a loss of these potential benefits, it can be fundamentally detrimental to hope; '*you're just like ah I'm back here, it's just the same old thing, why do I have to be here sort of thing' (20, YP)*. Effective relationships, in all their complexity and challenge, bring a deep emotional bond that can be lifesaving, giving '*you one more like reason to live and it would give it a bit more time so you think she's right, because she is going to be here for me, maybe she's telling the truth, maybe she will be' (17, YP)*.

**Theme two—Cultivating connection:** *'I think a relationship is about kind of having a connection' (4, Staff)*.

A mutual connection is part of the foundations of a therapeutic relationship, with important qualities and requisite ingredients. Although all parties input into this, staff have a crucial role to play in how they behave and approach work with young people and families:

*It's just a case of building a connection between two people and sort of being personal, like talking to staff about their families and their lives and stuff like that (12, YP)*

This personalisation is integral to encourage a deeper, more genuine connection, whereby increasingly intimate and sensitive discussion can occur, enhanced through the inclusion of safety, openness, boundaries and support; '*For [daughter] it's that she's kept safe, but also challenged' (16, Carer)*. This sense of safety establishes an environment of open, non-judgemental conversation, '*where you can talk to somebody about how you feel, like you can sob your heart to them, you can just tell them whatever is going on in your mind, you can just let it all out' (23, YP)*. Such openness is promoted through the actions of the clinician and ultimately, staff personality characteristics play a substantial role in opening communication channels that generates insight to young people's mental wellbeing, through accessing their thoughts and feelings.

Whilst staff have a crucial role in building this sense of connection, all parties are required to input; there is mutuality in its creation. This must be underpinned by shared respect and trust; '*there should be, that length of respect to anybody, as you know respect the next person as*

*how you want to be respected' (14, Staff)*. If this is not present, it poses a considerable barrier to treatment outcomes; *'if you don't have mutual respect, then you don't get the trust, if you don't have the trust, then the therapy is pointless' (1, YP)*. Mutual trust facilitates connection via the revealing of inner experiences:

> *I know that when I get on one to ones with staff its, where it's just me and the staff member and they can actually just sort of talk to me about how they feel about things on the ward and it's kind of creating a sense of trust, like I really trust those people and I feel like they trust me as well, that's definitely beneficial to me. (12, YP)*

Carers further reiterate the importance and yet difficulty of trust across all parties; highlighting its fragility, with failings impeding relationship building and increasing the likelihood of ruptures, necessitating genuine honesty and transparency from all involved:

> *we will share with [YP] what we spoke to the staff about, because at the time we start lying, you will always get caught, and especially if you've got a three way relationship, something is gonna catch you out, and the minute [daughter] catches me out, I've got an issue cos she goes you've lied to me, then I'm back in a different area, she doesn't trust me. (22, Carer)*

These more abstract relational ingredients that contribute to the connection are facilitated by activities that promote common interests, shared experience, casual and sometimes humorous conversation. Alongside staff members having the personal qualities to aid connection, they must also be given opportunity for this to occur, *'if you watch a movie with someone you have shared common interests, you make history together, something to discuss with' (9, Staff)*. Participants generally preferred such interaction to be more casual, as a more personal connection reduced young people's anxiety and encouraged more open and honest communication:

> *when it's a more casual setting with someone and you know they're a member of staff but you also feel a more personal connection because they have been normal with you . . . I find it a lot easier when I'm just talking to someone in a casual situation than if I'm sort of being interviewed, it's not that I lie, but it feels more natural and it sort of creates less anxiety. (12, YP)*

Whilst this casual context, and the wider cultivating of connection, was viewed as a joint endeavour, staff were highlighted as having a key role in fostering this, via actions, broad approaches and availability; *'You find something that they are interested in . . . you sit down, you chat . . . you find out how they like their tea, you take it to them, you smile, you just sort of find a way . . . spending time in the communal areas, it's the little things' (5, Staff)*. This suggests early cultivating connection does not necessarily require formal discussion or structured sessions, but rather a relatively normalised situation whereby two people invest time in being present with one another, and express an interest in what each has to say. The nature of young people's difficulties may pose an initial barrier to this connection, although staff participants state this can be overcome by minimising the interaction further; *'I just want to be here, I want to see, I just want us to be together, no talk, nothing. So we both sit, nothing, and it works' (9, Staff)*. Such a technique mitigates the need for verbal communication, and associated anxiety, whilst allowing the clinician to demonstrate they care about the young person's welfare, want any interaction to be on the shared terms, and are available if needed. These positive features were valued across participant groups; *'he's not intrusive or anything like that; he just lets you flow you know what I mean and listens and guides a bit and he's very good' (7, Carer)*. Moreover, insight was gained as to why actions of normality were so well received:

*It's a welcoming gesture, I don't know whether it's a [Northern] thing, oh your feeling sad I'll make you a cup of tea, it's that you know whole Coronation Street [North of England Soap Opera] type thing, we haven't got a pub on the corner but we've got a cup of tea. (13, Carer)*

Young people and carers value staff taking the initiative in building the connection:

*As much as we get told you don't need to be scared to ask and that you can just talk to us, we're not really gonna do that, we'd rather somebody came up to us and be like right we're having one to one time, how are you feeling and sit with us for a while and just talk with us and eventually just get it out of us . . . It makes you feel better knowing that somebody wants to do something to help you. (17, YP)*

Part of the staff role is also to be approachable, so that people can feel comfortable to raise concerns or initiate conversation if they feel able to; '*I thought if they're just approachable, I'd never come in and think I need to talk about that and I'm scared to say it, I do feel like if something is bothering me, they'll listen to me and you know it's not hard to say*' *(21, Carer)*. Being approachable is essential to cultivate a two-way connection, whereby all parties feel they have opportunity to contribute. This is augmented by all involved genuinely demonstrating interest in what is said, which is lacking when the young person is not acknowledged or given space to express; '*when it's not in-depth they're [staff] just like try and get out of it as quick as they can like oh yeah yeah I get it and all that, you know like skirt away, move on as quick as possible like they're not interested, you can tell when they do that*' *(23, YP)*. This can lead to considerable relationship ruptures; dissipating any feelings of being cared for, valued or listened to, and inhibit any benefits from a positive connection.

**Theme three—Knowledge is power:** '*you might say well that's [communication] overkill but actually this is our child, so nothing is overkill*' *(22, Carer)*.

Knowledge, understanding and information are important in the development of relationships. Recognising and sharing knowledge amongst staff, young people and families does more than communicate understanding; it recognises roles, reflecting and conferring mutual power and thus engendering meaningful collaboration. Where this does not work well, relationships can become strained. Participants agreed that it is important for all parties to develop an understanding of mental health generally; whether through personal and professional experiences or formal training:

*I think having a good theoretical knowledge is always useful as well, for what might be going on for someone, cos you can kind of, I know you shouldn't second guess what's going on for them, but it does help to frame it and think about. (19, Staff)*

*Their experience with patients, and I guess the training they've done, as well as other factors, like their childhood and stuff like that, affects everyone . . . they've learnt a broad range and then they cater to the one they're best with. (20, YP)*

Such increased understanding can be applied to young people's care, as well as demonstrating that, at least on a theoretical level, staff understand the psychological difficulty young people are experiencing. This provides a foundation upon which to develop more person-specific knowledge. Gaining insight to people's unique story and the nature of inpatient care is deemed important across participant groups, in order to build a trusting and supportive relationship:

*I think if you come onto the ward, you've got to spend quite a bit of time observing, getting to know that person, see how they are, have a look at them, watch them, understand what they do when they're not trying to show you what they do. (22, Carer)*

*Just learning by effect and cause like what that individual, what triggers them to be anxious, because they need to know that to work with them. . . they need to understand that and make an individual plan because not everyone is the same. (10, YP)*

Whilst this individualised approach to knowledge generation can support person-centred treatment, staff members need to carefully balance their professional knowledge with that they can garner in the context of the unique relationship with each young person. There is a risk of invalidating young people's experiences and relationship rupture, especially if a dichotomy exists between the young person's and clinician's treatment strategy. This may be augmented if the young person does not fully understand, or is unable to communicate, their mental health difficulties and needs. In addition, staff's understanding of young people must also consider them and their family in their cultural context; '*there's an Asian girl on this ward*, *who understood about festivals*, *it's been nice*, *they understand' (18, Carer)*. If there is a lack of cultural understanding or acknowledgement by staff, it is likely to lead to a disconnect in the relationship, as the young person or carer feels devalued:

*You're so shut off, you don't feel very valued, and you don't want to really bother with them, because you're thinking they don't understand you and that's it. (18, Carer)*

Moreover, knowledge is not static and needs to be shared and held in relationships as well as within clinicians, requiring a willingness for all parties to engage with one another to expand their mutual understanding. This creates a culture whereby professionals can share knowledge to provide support. However, carers experience blocks to this knowledge transfer:

*It's a bit like being a detective because the onus is a bit on you to ask the questions, because the questions don't get addressed*, *if a parent was not equipped to ask those personal questions then that information wouldn't necessarily be forthcoming. (13, Carer)*

Proactively using effective communication to distribute information is crucial and where this does not happen effectively, it can impact on relationships, experiences of the admission, and treatment outcomes. When knowledge is not shared in a transparent and accessible way, individuals are likely to feel disempowered; '*more transparency [is needed] instead of all these hidden laws*, *because they like to say that there's a lot going on behind closed doors' (20, YP)*. Transparency, coupled with being proactive is essential to ensure that family members feel valued and involved:

*It's communication cos there is a total lack of communication and interaction with the parents or carer, we should have a debrief of what's happened, not saying every day, but even once a week, we could sit down with the named nurse once a week and you might say that's overkill but actually this is our child, so nothing is overkill. (22, Carer)*

Carers also have considerable knowledge to share with staff, and if this is not appropriately acknowledged, it can be invalidating; likely placing a considerable strain on the relationship:

*I collated thirty odd pages of all the things and events and thoughts and why, why she [daughter] operated like she did, and I sent it, and it was well have you got it, oh I don't know, and it was highly confidential stuff, but there was no acknowledgement that it was received . . . they did receive it in the end and it was read, and they said it was useful (13, Carer).*

It is necessary for staff to build positive relationships with carers in order to value and share this knowledge, which in turn will further foster a positive relationship, so as to inform their individualised treatment strategy; *'just communicating it in a way that's positive and making sure that she understands and that she still feels safe and that the trust's still there' (3, Staff)*.

**Theme four—Being Human:** *'human support; they are listening to my breathing and I am listening to yours, and that is it' (9, Staff)*.

Alongside their roles in the system (staff, young person, carer), all parties bring their human selves to the relationship. It is crucial that individuals are treated as such, with their unique qualities, perceptions and needs. This is difficult for all and can have specific consequences for staff who face additional emotional challenges as a result of this authentic vulnerability, balancing this humanness with their nursing role, moving along the spectrum between the two as required:

> *End of the day, they're people and we're people . . . and that needs to be something that we need to spend a bit more time on I think. (5, Staff)*

Participants all recognised that young people are more than patients or even 'young people'; they are human beings with individual differences and needs. They bring with them prior experiences, expectations and qualities that need to be recognised and worked with in order to develop effective relationships; *'every child is different and every family is different, there's no handbook, it's the same with having kids, none of them are the same' (3, Staff)*. When this individuality is not recognised, it is experienced as painfully dismissive and rejecting, *'I'm just another number that just needs to get shipped out of here' (6, YP)*. Perhaps surprisingly, the inpatient environment can also be a space where young people are able to connect with their identity; *'to sort of just be themselves which they never really had an opportunity to do' (2, Staff)*. This may be due to the creation of a safe emotional space that encourages open communication, whereby mental health difficulties are destigmatised and young people no longer need to 'hide' this part of their self; allowing a freedom to fully be, and develop, themselves. In relation to this, young people and carers recognise that they have specific needs that others might not share and that might affect the relationship:

> *I'm one of these people I get attached to things and people way too easy, and I know it's not the case with everybody but I do. (17, YP)*

> *[Son] is not a child that would go up to someone and say he needs help, he needs prompting all the time . . . I have felt that some staff have just left him. (24, Carer)*

Family members and carers are also human beings with individual differences and emotional needs; it is important these are recognised and supported in order to foster therapeutic relationships:

> *I've cried in sessions . . . if there is a parent sat there, who is bringing her daughter or son to appointments week after week, year after year, they're invested and they care, but they're suffering as well, I don't think it's rocket science to realise that, to acknowledge it. (13, Carer)*

This encapsulates the emotional hardship parents can experience, providing clear evidence for the need for staff to provide support; especially considering the parent or carer is often the primary supportive figure to the young person, and thus an important protective factor. Young people too acknowledge that their parents are people in themselves who might need support from staff and welcome that; *'I'm grateful that the staff here have actually, they're giving*

*support to her [mother] as well' (6, YP)*, emphasising the triumvirate nature of the relationship. In response, staff have to carefully assess and react to individual family member's needs, being attuned to their preferences. Participants felt such individual differences could be challenging, exacerbated by the different roles that members of the same family might occupy:

> *I follow what the nurses have told me, whereas [mother] she doesn't, she's like what have you done [self-harm] with, how have you done that, where's that come from, what caused you to do that . . . because she's her mum that's what she's gonna do. (11, Carer)*

Such variation across and within families increases the potential for communication difficulties and associated distress, emphasising the need to ensure opportunities are provided to explore different family members' needs and perceptions. As healthcare professionals, it is staff who are best placed to not only facilitate this, but genuinely acknowledge and appropriately respond to such considerations.

Staff members are people too and have to bring their authentic 'human selves' to work in order to cultivate therapeutic relationships. They also face a unique challenge in having to balance this with occupying nursing roles and professionalism, balancing both dynamically; *'they can have a really good time with you, but they know that you're here for business and they know that you keep them safe' (3, Staff)*. This requires fluidly shifting between positions, depending on the circumstances:

> *You have to give a bit of yourself to it, for it to be a really honest relationship and to actually change anything, you've got to put something of you in there that isn't just the nurse, it's you, and that goes against the nursing side of it. So it's really, it is wearing two hats. (19, Staff)*

However, staff member's humanness comes with vulnerability; an exposing of the self that fosters the relationship and can also, crucially, support young people to sit with their own vulnerability:

> *we are human beings, we're not robots that can do the perfect thing every single time, and it's about recognising that in themselves as well. So a lot of our patients hold themselves up to a much higher standard to other people, and when it doesn't go that way for them, they start getting distressed about it, and it's just knowing that yeah we're people, and just do your best with what you can, that's all anyone can do. (15, Staff)*

Acknowledging and projecting this humanness can model a more realistic sense of self to the young people, potentially lessening the disproportionate expectations they place on themselves, and the. Although, being human can be at odds with 'doing the job', when the job involves so many tasks:

> *I think they need the right skills of how to approach people and as much as professionalism is important. . .. they need to let a bit of a human side come through and smile and be kind and make people feel comfortable rather than just doing the to do list every day. (6, YP)*

Consequently, staff members have to step away from this professional role sometimes, which can reap benefits for the relationship; *'I would say they would need to be able to switch off from nurse mode at times' (12, YP)*. Talking to a 'human being' as opposed to a 'nurse' not only promotes better engagement and communication, but also develop the young person's sense of self:

*It's nice to have conversations with people and it's nice that they see you as normal people as well- that's a big part of it. When they can sit and have a conversation with you it's enforcing that idea that you're not just a mental patient on a mental ward. You're a person. (12, YP)*

If a young person is able to make this distinction, it is likely to have a profound effect on their perception of themselves, their future following discharge, their treatment and their recovery. As such, modelling, promoting and developing humanness is essential for positive recovery outcomes. Demonstrating vulnerability and expressing emotions is a way to show this; actions that young people highly valued. Aside from the therapeutic benefits such compassionate actions can have, they also allow staff to be seen as flawed, available and distinctly human:

*The staff here are some of the best staff you will find, because they're not the whole scary nurse or whatever, they have depth and they're genuine people, and so I think seeing the nurses as their own complex and their own people, rather than robot people. (1, YP)*

However, staff cannot simply, 'be human' and bring their inner selves, they also have to balance this with professionalism, with an ability to bring distance from their inner world. Balancing this sense of human, open connection and formal nursing responsibilities was suggested by participants to be managed through the application of boundaried professional care:

*[nurse] is quite a direct person, and she has to be a figure of authority, and [YP] doesn't like authority, so she sees [nurse] as that person, but then other times [nurse] will come in and she's having a chat, she's happy laughing and joking, she can get on with her, and she knows that [nurse] can take it as much as she can give it, so she has a slight, I suppose it's a love hate relationship, but she's quite fond of her in a way. (22, Carer)*

This exemplifies how a staff member can effectively balance their personal humanness in cultivating a positive relationship, with the necessary authority to implement professional boundaries and provide containment. However, moving along this human-professional spectrum brings costs as well as benefits, as staff can experience impacts on their emotions, sense of self, exacerbated by the inpatient environment and acuity of young people's mental health presentations. These impacts can affect nursing staff in a way that may impede their ability to move towards the human end of the spectrum and apply their positive personality characteristics, '*[staff] would be really stressed, and there is only so much you can be yourself when you're tired' (1, YP)*. Such outcomes inevitably lead to disengagement from the relationship, hindering potential benefits, and possibly exacerbating the emotional distress young people experience. Consequently, staff have to be able to let difficult emotions wash over them:

*I take it like egg you know where you boil egg the water doesn't stick to the shell of the egg, then when you remove the shell, the egg from the water dries up. That's how I see the relationship here, like both with young people or parents, not to take it to heart. (9, Staff)*

Here, the staff member describes how when boiling an egg in its shell if you remove it from the water, the surface dries immediately. Therefore, this idiom illustrates that staff need to let things wash over them quickly–be impervious to emotional distress. Another version might be 'like water off a duck's back'. In reality, this cannot always be possible, while also being vulnerable and human and open. Coping by cutting off is a short-term solution but comes with costs–the vulnerability, connection and care that young people and families value so much; '*I*

*think it is something people struggle with. . . I think a lot of nurses manage by not letting them-selves care too much, or be too involved' (19, Staff)*. This not only highlights the difficulty of bal-ancing humanness and professionalism along a fluid spectrum, across countless interactions and circumstances, but also the potential danger of staff positioning themselves too firmly at the professional end in order to protect their own emotional wellbeing, when not offered any alternative support to manage this emotional pain.

**Theme five—The Dance:** '*you're almost just taking that hat off and putting another one on' (19, Staff)*.

Therapeutic relationships in inpatient CAMHS are like a dance. They involve multiple pro-tagonists, with different positions, moving with fluidity across space and time. Young people, carers and staff have to occupy different positions and roles over the course of the admission, playing different parts and coming together or moving apart. Staff members have to be partic-ularly flexible in order to facilitate this and, alongside the human-staff spectrum noted above, have to shift in other ways. Roles and positions come with responsibility and power and although we might seek equal collaboration among all groups, there needs to be flexibility in this over the course of the admission.

Being part of the relationship- having a position in that—helps people feel involved and yet is also taxing. Staff members seek to acknowledge young people and family members in a way that recognises their position and humanness. Carers value staff taking on some of their role for a while:

> *We are relieved at that, we're happy for that because somebody has, not taken our responsibil-ity away from us, but saying to all of us, this isn't right, this isn't safe . . . maybe it's us stepping back from responsibility a little bit. (16, Carer)*

Whilst this brief 'respite' is well received, not just for carers emotional wellbeing, but to pause and review the system that may influence a young person's mental health, there is also a concern of staff taking on too much responsibility and negatively impacting young people's sense of agency:

> *Quite a lot of dependence [in the relationship], if you let that happen, it's almost like you're responsible for all their feelings and thoughts. There is a lot of expectation for us to manage everyone's distress, and they almost stop trying you know cos we're here to do it for them, so I think in some ways they are deskilled being here for a long time and become really dependent on particular people they've got good relationships with. (19, Staff)*

This taking on of responsibility can exacerbate the emotional burden that staff experience. In addition, there's a fear that it could render young people overly reliant on others, at risk of becoming deskilled or disempowered and limiting the normative adolescent development towards independence. Staff members have to navigate these complexities, balancing the need for care and independence, for safety and empowerment. These concerns are not confined to staff and parents, but young people similarly want agency in the interactions and decision making, although this can contrast to the inpatient setting whereby '*you've got no control over what's happening. You can't change anything that's happening. . . No, you just have to sit there and let everything happen. . . [its] not nice cos' you wanna' be in control' (8, YP)*. Whilst this likely negatively impacts ownership over their recovery and prompts disengagement, partici-pants equally stated they value staff that recognise their position and share power in the rela-tionship; '*you're not pushed to do anything, you're encouraged . . . there's no obligation or anything like that, it's very much on your terms' (1, YP)*. Empowering young people in this way

helps them have confidence in their own abilities establishes a safe culture of growth, with a supportive structure, that allows young people to develop their skills at their own pace.

The triumvirate nature of the relationship requires particular recognition, responsiveness and flex, with a need for staff to occupy some of a parental position, without undermining their role; *'you're guarding them aren't you really. . . but it's not a triangle' (16, Carer)*. Such positioning can be challenging for parents, who as the party not based on the ward, may already feel the triangle is unequal and their position threatened. Young people were also aware of this; *'when I've had problems in the past she'll [mother] just say oh well they think they know what they're doing so you need to rely on them and I'm not sure if she feels like she's getting edged out, by their input' (6, YP)*. This places considerable onus on staff members to provide opportunities for involvement flexibly, which doesn't always happen:

> *I'm a bit pushy and want to know anyway, so I sort of put myself out there and I think I've developed relationships where they kind of know me and yeah. . ... as a parent it's really difficult to just sit back and well. . ..something's gone on there, can't ask too much, you know so you feel alienated in that respect.(13, Carer)*

Subsequently, staff, young people and carers all have roles–parts to play in the dance of the therapeutic relationship and although all vital, these can vary. Some see their role in the relationship as balanced with that of staff; *'like fifty-fifty. . . because you have to put some effort in' (8, YP)*, but with staff bringing their particular skills and taking some lead; *'it takes both people to start a therapeutic relationship, but obviously one's only gonna be able to provide the therapeutic side, but professionalism from the staff, they know when to approach you first and so that quite helps (20, YP)*.

This highlights how everyone is involved in the dance, but it is nursing staff who must proactively take the lead and apply professional skills to establish a therapeutic framework that can develop the skills of the other parties so that they may have a greater role.

During the course of an admission, there is a need for flexibility and movement between and roles, as needs arise and skills are developed. Staff have to be responsive to the movements of others and the emotional tone in the context; *'You could come in one day and it's a really nice sort of day, the young people are all getting on with their own things. But you could come in two days later and every patient is sort of struggling with different things' (2, Staff)*. Such fluctuation creates a sense that staff have to flex to the moment-to-moment needs of young people, which they too acknowledge:

> *There's been times where I've pushed people away. . . when I've felt I've been really in need of help and support.. . .and I've not like been in the right mind set to get on with people. (10, YP)*

This exemplifies how mental health difficulties, coupled with difficulties forming positive relational connections, can prevent young people from both eliciting and accepting support, despite having awareness that it will be beneficial. Considering this and the need for flexibility, staff must maintain a sense of continuity; keeping the rhythm of the dance even though the steps might change:

> *Being consistent, constantly showing that you are available at different times if they need you and if they don't, that's fine. If they want to just sit and have a brew, that's fine, but if they want to sit and have a conversation about how rubbish their day has been, that's also ok. And just sort of constantly being there really. (2, Staff)*

Working in this way not only responds to the variation of young people's care needs, but also continually reassures them support is available whenever they feel ready to accept it. However, staff acknowledge how this shifting and changing of roles can be difficult for young people and challenging for the relationship, despite its necessity:

*You know it's really strange cos I've had a really good relationship with somebody where they've talked and bared their soul to me, and the next half an hour I'm holding them for a feed and that's really hard to manage, and I think it's hard for the patients to manage as well, cos you're almost just taking that hat off and putting another one on. (19, Staff)*

The need for responsiveness can be particularly apparent when approaching discharge; when the relationships with the ward are ending, fostering autonomy and connections with community. Staff recognise the anxiety that this period brings for young people; '*once they start looking at discharge, they think well how am I gonna make it on the outside, who am I gonna have, am I gonna have these people, am I still gonna be friends with the peers that are on the ward with' (15, Staff)*. Also, young people see how their relational journeys have only just begun when they leave the ward:

*A lot of the time I think the problems in family relationships when you're in here happen because your family couldn't support you on the outside cos' they didn't understand or couldn't, didn't have the ability to, and then when they come in here you might get out but there's still, it's all about when you come back out. And how they're gonna be able to support you when you come back out. And I think that's very important. (6, YP)*

Such concerns highlight the importance of staff developing the skills of other parties in 'the dance', so that it can continue without them post-discharge. This is especially true for carers, who play a vital role; '*you kind of got to be doing it together haven't you, because your gonna live with them, your gonna take them home aren't you' (16, Carer)*. If they are not equipped to manage this, it could bring future mental health deterioration and re-admission.

**Theme six—It's tough for all of us in here:** '*there's never too long without some sort of kerfuffle' (12, YP)*.

Inpatient CAMHS units–the systems in which these relationships exist—are challenging to be in for young people, family members and staff. This challenge is somewhat–though not fully—shared and mutually recognised. Managing these challenges necessitates both individual and systemic solutions.

Young people are often bored, there are arguments and disagreements; staff take time to manage this and this in turn can detract from building relationships, further impacting ward atmosphere:

*I spend a lot more time in my room because I don't want to get in the way and because staff are sort of more busy there's less staff around, it's a lot more likely for even more arguments to kick off . . . part of why these guys are stressed out is because they are constantly trying to stop arguments from happening . . . It feels like there's never too long without some sort of kerfuffle; the social politics of this place that I don't want to be a part of anymore- like I give up. But it's just stressful because you know one minute you think everything's alright and then next minute two people are having a go at each other (12, YP)*

Such difficult circumstances may lead to increased isolation and hinder positive connections, coupled with ward procedures that can take time away from focusing on the

relationships; *'it's the relationship between the patient and nurse that really matter, not what's behind an office, cos you have a lot of paperwork to do, so maybe letting time to have a break from the paperwork, that could help build relationships' (20, YP)*. Staffing levels impact the basic provision of opportunities to cultivate connections and build relationships. These systemic concerns may create a sense that staff do not have time for young people, fostering feelings of invalidation and impacting trust and connection.

Some of these systemic challenges are mutually recognised by young people, staff and family members. Staff members recognise that inpatient environments are not welcoming; *'it could be the first time ever away from their mum and dad or their carers, their first maybe ever away from home, their brothers and sisters. And it's a place they've never been before, people they've never met and quite sometimes a loud noise environment' (2, Staff)*. Also that the nature of the admission can make it difficult for them to build up relationships; *'there is a pressure of time, because they're not going to be here forever, and if that's hard for us, then it's going to be super hard for them' (5, Staff)*.

Young people see the emotional toll that the environment takes on staff; *'Staff have been in tears because people aren't making their jobs very easy. I know some staff having to take days off because it's just too much working here' (12, YP)*, and, in turn, feel the impact themselves; *'when things are difficult on the ward it really sort of shows in the staff as much as they try and hide it . . .we're not a stupid group of kids' (12, YP)*. Carers also see the strain; *'we understand that's not always [easy], you've got alarms going off, we're not daft you know, we know' (16, Carer)*. The emotional toll of the job can act as a barrier to staff members moving towards the 'human' end of the human-staff spectrum and thus hinder the benefits this can provide. This potential shortcoming is recognised by family members who value nursing staff and their skills in particular, yet acknowledge how the systems they are part of can impede the core of their professional remit and cast a shadow on their values:

> *Nurses are very important people and they work hard, and they're not valued much, and I think they shouldn't let, I know it's hard, the pressures of their work, cloud their relationship with the patient or their family . . .they've gotta do their job, but to keep an open mind. (18, Carer)*

Although many of the challenges of inpatient wards are shared, some staff members experience racism, and feel alone with this; *'Could be just racist, it boils down to that I would think and I mean a lot of agency staff come here, they are black, so there is always that stigma between you know agency staff so I dunno' (14, Staff)*. In addition, staff who identify as black or mixed race experience a disconnect from some young people and struggle to make sense of its origins:

> *I don't know if it's a cultural thing and they think we are immigrants, like we're black people and they think we are not from this country and we don't understand certain things . . ... I don't know if it's just a young person thing but some of them do that to you. (14, Staff)*

These staff members draw on personal resources to help manage this particular challenge and work towards an alliance; *'my ethnicity plays along with trust like I felt even when I want to establish a relationship they draw back because of my ethnicity . . . so to build that relationship takes extra like strength or extra input from me before the trust can be established' (9, Staff)*. However they need specific support from the system that is currently lacking:

> *As a coloured person I feel like I'm always excluded from activities on the ward, like going to cinema, I've never been to cinema, up to now I have never had the kind of ride in the car with*

*them, they don't put me on one to ones to go on the walk on the grounds, but it's changing a little bit now . . . So the management needs to do something about that. (9, Staff)*

Mitigating the threat from the system on the development and maintenance of therapeutic relationships involves influencing the ward atmosphere, reaping the benefit of positive social interactions and support, and utilising personal qualities or actions. Having spaces to understand and resolve issues related to the alliance are helpful yet hard to access if not prioritised; '*we've got a group ran by one of the psychologists to sit and talk about any problems, and that's really good and really useful, except that we can never go because there is never enough [staff] on the ward, there is always this or that going on*' *(19, Staff)*. Carers also acknowledge that staff will need specific forums for emotional processing and learning in order to foster their relational skills with young people, and, like staff, want this to be prioritised; '*some kind of supervision where they can talk through things that have happened, I don't think there should be a compromise on that*' *(16, Carer)*. Such supportive measure for staff have to be prioritised in order to ensure their emotional wellbeing is sufficient to allow them to bring the positive aspects of their 'human' selves to the role.

Staff members rely heavily on one another for support, in the face of systemic challenges:

*We've got a great team, always like backing and when you struggle with anything they are ready to explain to you, not victimising, not criticising. (9, Staff)*

Whilst this can provide a robust support network that can allow good clinical practice to become further embedded within teams, if staff are under increased pressure and at higher risk of burnout, this may also become embedded. Similarly, when it's not accessible, wanted or effective from staff members, young people and carers have to seek support from their peers, in order to manage the system; '*sometimes we don't find it easy to talk to staff and find it easier to talk to each other, and I know we all have our own problems but honestly, sometimes talking to each other is what helps more than anything, cos we understand each other*' *(17, YP)*. This can help foster a sense of togetherness and understanding; '*I met one of the other patient's mums . . . and I've since developed this friendship with her and that's been really helpful because you have somebody to talk to about the same experiences and how you've felt*' *(13, Carer)*. Finally, sometimes, sharing a laugh is the best medicine, for the predicament all find themselves in; '*I feel like humour is a lot of people's way of escaping that*' *(1, YP)*.

## Discussion

In the context of the crucial importance and endemic challenges of inpatient mental health care for young people [5] this study aimed to explore the experiences of young people, nursing staff and carers in this setting in order to develop a rich and comprehensive understanding to inform recruitment, training, support, policy, and service development initiatives. We focused on therapeutic relationships, given their complexity, necessity and impact in mental health care [12] and particular barriers that might be faced in adolescent inpatient settings [17].

Our findings confirmed that therapeutic relationships within inpatient CAMHS are essential, complicated and unique in their contents, process and impact. They require immense skill, personal qualities and understanding in order to cultivate reciprocal connections between parties that balance consistency and flexibility. We present six themes that were present across participant groupings and represent the nature of therapeutic relationships in this context, their impact and both the facilitators and barriers to their effective development, maintenance and implementation. Therapeutic relationships were seen as key: *they are the treatment*. All

parties have to work on *cultivating connection* through openness, activities and conversations. Staff members must combine their professional expertise with a thorough understanding of the young person and family's specific histories and needs as *knowledge is power*. Young people, carers and staff bring their human selves to bear on the relationships, a process which comes with huge costs and benefits; nurses walking a tightrope between their personal and professional identities; *being human* is key [20]. The relationships and people's roles and positions within them shift over time and space, necessitating flexibility, particularly from staff, and a responsiveness and attunement that demonstrates immense skill and resilience; all parties enter into *the dance*. All this occurs in an environment that provides containment and yet untold drains on the very resources that are needed for this work; compassion, time, vulnerability, space [30, 31]. Within inpatient settings, *it's tough for* everyone and the negative impact of this context on relationships must be mitigated, particularly for those for whom it presents even greater challenges. That these relationships persist through this is testament to their durability and yet the importance of effective support for them cannot be overstated [34].

Participants talked of the need for personal qualities, reflection, training, information and support being shared amongst teams and individuals. The need for individuals' unique humanness to be acknowledged, alongside movement within the relationship reflects recent research in other inpatient settings [48]. The value of cultivating connections through 'normal' interactions also chimes with a recent themed review in this setting [17]. Systemic structures currently seem to act as barriers to this rather than a facilitators and it is evident that this can result in damage to the relationship and therefore harm to individuals and groups.

The convergence of themes (alongside the diversity of individual views) across young people, carers and nursing staff reflects the triumvirate nature of the experience [29] and the value in synthesising different perspectives along a common thread. Future research could address the limitations of the current study by synthesising the experiences of additional professional groups alongside nursing staff and ensuring better representation of diversity for all groups in terms of gender, ethnicity and other marginalised identities.

The analysis indicates that young people, families and nursing staff would be better served by a system that prioritises the formation and maintenance of effective therapeutic relationships. This requires adequate staff numbers, training and time in cultivating connection and doing 'normal' things together. Consideration should also be given to aspects of the workforce that might impact on this being successful, such as the role of agency or bank staff, where continuity of care and relationships might be impeded. Staff members need support to explore and understand individual histories and needs alongside clinical information. The balance between being human and professional is a tricky one and would benefit from 'live' focused relational and emotional support alongside more static training and supervision. Service structures, protocols, care plans and training also need to reflect that therapeutic relationships are inherently multi-layered and require a responsiveness and flexibility across time, space and people, alongside a commitment to consistent values-based practice. Admissions, transitions and endings might be points of particular vulnerability and so paying attention to relationships at these times will be key. Mitigating the impact of the inpatient context of therapeutic relationships will require over-arching policies and structures, including those related to staff burnout, the inpatient environment, racism and staffing models. More generally, the themes presented here offer a framework for staff competencies in relation to building relationships that could form part of recruitment, training, supervision and workforce planning. We hope that the lived experienced testimony, synthesis and analysis shared here can serve to drive policy makers, service managers and clinicians to focus on therapeutic relationships, as essential to quality inpatient care, and afford them the structures, support and significance they deserve.

## Acknowledgments

We would like to thank the participants that shared their experiences with generosity and passion, and to the young people, staff and carers who remind us what is important and of the need for change. We welcomed design advice from Professors Penny Bee and Karina Lovell and the Moving Forward Young People and Carers' Council. We are grateful to Lisa Manning for her design input and the following transcribers: Nicole Wynford-Thomas, Rachael McNally, Naomi Shenton, Ellie Young and Hannah Simpson.

## Author Contributions

**Conceptualization:** Samantha Hartley, Katherine Berry.

**Data curation:** Samantha Hartley.

**Formal analysis:** Samantha Hartley, Tomos Redmond.

**Funding acquisition:** Samantha Hartley.

**Investigation:** Samantha Hartley.

**Methodology:** Samantha Hartley, Katherine Berry.

**Project administration:** Samantha Hartley.

**Supervision:** Samantha Hartley, Katherine Berry.

**Validation:** Samantha Hartley, Katherine Berry.

**Writing – original draft:** Samantha Hartley, Tomos Redmond, Katherine Berry.

**Writing – review & editing:** Samantha Hartley, Tomos Redmond, Katherine Berry.

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
