## [Decision Letter · Decision Letter 0]

14 Jul 2021

PONE-D-20-31788

Therapeutic relationships within child and adolescent mental health inpatient services: a qualitative exploration of the experiences of young people, family members and nursing staff

PLOS ONE

Dear Dr. Hartley,

Thank you for submitting your manuscript to PLOS ONE. After careful consideration, we feel that it has merit but does not fully meet PLOS ONE’s publication criteria as it currently stands. Therefore, we invite you to submit a revised version of the manuscript that addresses the points raised during the review process.

The manuscript has been evaluated by three reviewers, and their comments are available below.

The reviewers have raised a number of major concerns. In particular, all reviewers note a similar concern that the manuscript requires significant revision to improve the overall flow and structuring of the manuscript, paying particular attention stating the rationale and refining the focus further. In addition, they request further details in the methodology section, such as greater explication of the coding process, as well as shortening the Results section.

Could you please carefully revise the manuscript to address all comments raised?

We look forward to receiving your revised manuscript.

Kind regards,

Avanti Dey, PhD

Staff Editor

PLOS ONE

Journal Requirements:

5. Please ensure that you refer to Figures 1, 2 in your text as, if accepted, production will need this references to link the reader to the figures.

6. We note you have included a table to which you do not refer in the text of your manuscript. Please ensure that you refer to Table 1 in your text; if accepted, production will need this reference to link the reader to the Table.

Reviewers' comments:

Reviewer's Responses to Questions

**Comments to the Author**

1. Is the manuscript technically sound, and do the data support the conclusions?

Reviewer #1: Yes

Reviewer #2: Yes

Reviewer #3: Partly

2. Has the statistical analysis been performed appropriately and rigorously? 

Reviewer #1: N/A

Reviewer #2: Yes

Reviewer #3: N/A

3. Have the authors made all data underlying the findings in their manuscript fully available?

Reviewer #1: No

Reviewer #2: No

Reviewer #3: No

4. Is the manuscript presented in an intelligible fashion and written in standard English?

Reviewer #1: Yes

Reviewer #2: Yes

Reviewer #3: Yes

5. Review Comments to the Author

Reviewer #1: I very much enjoyed reading this interesting and timely paper. The approach is well-judged for the topic and the analysis shows reflexivity and sensitivity. My suggestions for improvement are as follows.

ABSTRACT

• The abstract should focus more on describing the method and results of the study (if wordcount is an issue, the first 4 sentences contain redundancy and could be condensed into 1-2 sentences). The breakdown of N=24 across participant groups should be clarified.

INTRODUCTION

• The Introduction seems to be specific to the UK context (e.g. 2nd sentence references NHS commitment to improved access to community services) – this should be stated at the very start.

• Line 65: “manage the ‘push and pull’ of working with adolescents” – meaning unclear

• Page 3 states “there has been no in-depth study reporting on the alliance in this context from the perspectives of all key stakeholders.” This is somewhat unclear to me. Do the authors mean that no previous research has taken a qualitative approach to explore the therapeutic alliance? Or simply that no studies have involved multiple participant groups in the same analysis? If the former, this should be more clearly stated. If the latter, the authors should explain why combining multiple groups into the same analysis is important or valuable, above and beyond a collection of individual studies that each focus on one stakeholder group. Additionally, if previous studies on the therapeutic alliance have explored the unique first-person perspectives of young people/staff/carers, the key findings of such research should be outlined in the Introduction.

• The Introduction should do more to articulate the rationale for a qualitative approach to this topic, and for a critical realist epistemology specifically.

METHOD

• Were the participants related to each other? E.g. were the young people and carers from the same family, or did the staff provide care to the specific young people interviewed?

• The coding process should be more explicitly reported. The statement that analysis was performed “utilising paper, Word and Excel” raises questions about how systematic/consistent the process was. The coding frame should be described e.g. with number, hierarchical structure, examples of codes. Were all participant groups coded using the same coding frame? Was coding conducted by line, paragraph or purposively selected sections of text? How were codes developed into themes? What steps were taken to ensure the credibility of the analysis?

• Table 1 – please also include age ranges

RESULTS

• The Results show excellent analytic insight and the figures are very helpful in navigating the complex content. However, this section is very long and I think the authors should consider streamlining it. It is demanding a lot of readers to digest this quantity of complex material. The length means that important findings are getting ‘lost in the crowd’: after finishing the paper, I was not left with any solid sense of what its specific core contributions were.

• Line 587-590 – sentence long and difficult to follow

DISCUSSION

• The Discussion needs further development. It should include (a) a concise summary of the key take-home points of the research, (b) more explicit articulation of what the findings add to existing literature i.e. how they reinforce/contradict/expand on previous research, (c) more consideration of the real-world implications of the research (e.g. the importance of providing staffing resources that acknowledge the therapeutic importance of casual ‘down-time’ and staff emotional support), (d) critical reflection on the strengths and limitations of the study, with suggestions for future research.

• I was somewhat surprised to see no mention of continuity of care in the Intro, Results or Discussion. If the therapeutic relationship is so important, presumably it is important that young people are dealing with the same staff on an ongoing basis. Of course this is less an issue in inpatient than community services. But there is mention of agency staff in the Results – would these people just enter the unit on a very temporary basis and how does that impact on relationships? Might the Discussion consider any implications for continuity of care/facilitating ongoing contact with staff after young people leave the inpatient unit?

Reviewer #2: Summary

This paper presents a qualitative exploration of the role of therapeutic relationships in Child and Adolescent Mental health Inpatient Services from the perspective of service users, their family/carers and the professionals who provide these services. The contribution to the field is welcome and timely given the increasing demand for these services, which is anticipated to be exacerbated by the Covid-19 restrictions. By better understanding the importance of the role of therapeutic relationships, service delivery can be improved which could result in better outcomes for the young people involved, thereby improving their wellbeing and future opportunities in adulthood. The paper therefore has considerable practice and policy implications. The insight and time provided by these young people, their parents/carers and nursing staff is invaluable, and it would be unethical not to publish such findings. There is also a strong argument for the paper backed up by relevant literature, and the themes identified are enlightening and supported by the data presented. However, the manuscript is lacking some basic elements which would expected from an academic journal article. Therefore, I suggest a few revisions and encourage the authors to re-submit.

Major issues

Findings: You need to start the findings section outlining what your 6 key themes are and the order in which you will present them (and reference fig. 1). Then you need a small section on participant characteristics with reference to Table 1.

Discussion: Missing implications for future research and you need to be more specific about the implications for policy and practice. You could provide recommendations.

Discussion: Missing study strengths and limitations section.

Minor issues

Line 102 under “Sample”: It reads as though you only interviewed 8 YP, carers and nurses in total. Make it clear that you interviewed 8 of each and include the total sample size (24).

Line 127: How did you meaningfully engage the young people in developing the topic guide?

Line 142- 145: Your reflection on the first author’s relationship with participants is good, but could you provide an example regarding your reference to inpatient wards, relevant to the study, and explain how you carefully considered this?

Line 172: Explicitly state this is the first theme in the subtitle e.g. “Theme 1: Therapeutic Relationships are the Treatment”

Quotes: Number the participants so the reader can see if it is the same or a different participant with a certain view (e.g. YP1, Staff1 etc.).

Figures and tables are not referenced in the text- please reference and discuss them.

Table 1 is unclear and needs formatting, e.g. the headings for carer group, staff carer group, young person group are not aligned. You use (n=x) for some totals but (x) for others. I think it would be clearer if you had the participant groups as columns at the top and characteristics as rows down the side. If some groups do not share a characteristic, leave it blank. It is very difficult to read.

Reviewer #3: This is a generally well-written paper which covers several topics around child and adolescent mental health inpatient services. One of the major strengths is the inclusion of interviews with staff members, young people using the service, and carers or family members. This has allowed for a more well-rounded understanding of the topic and the authors have brought in multiple perspectives to illustrate each theme.

However, the structure of the paper lets it down somewhat. There is very little sense of coherence or how the themes inter-relate, and I think is partly because there are a large number of themes and subthemes, some of which don’t necessarily seem to come from the data. The themes are not even listed in the abstract, which makes it difficult to get a sense of the main argument of the paper. Each theme has a lot of ideas contained in it and it is not always clear how they are related – and some quotations used don’t seem to illustrate the point that is being made.

The themes all have sub-themes, which is helpful in understanding the data – but they are very generic/vague. For example, a sub-theme of the last theme is “what makes it hard” – someone just looking at the theme structure would have no idea what that actually meant. Instead, you could name the subthemes more specifically (e.g, “racism”, “boredom”, “emotional toll on staff”) so it is clear what your themes relate to.

A few of the themes did not seem to have their genesis in the data –for example, the metaphor of “the dance” did not seem to be mentioned by any participant and appeared to be superimposed by the authors, but this wasn’t explicitly acknowledged. (Or if this came from elsewhere – for example, if there was a theoretical basis for using this term - it would be helpful to acknowledge that as well.) Also, it just seems a strange metaphor to use if it didn’t come from the data – “Young people, carers and staff have to occupy different positions and roles over the course of the admission; playing different parts” – do people necessarily play different roles and parts during the course of a dance? Also the topics explored in these theme don’t seem to have much to do with the name “dance” (e.g., the topic of responsibility etc). You do refer to the idea of a “triangle” which DID come directly from the data – perhaps this would be a better theme?

Similarly, the theme “Knowledge is Power” is a bit of a strange summing up of the qualitative quotes in that section as very few of them actually speak of power, and “knowledge” seems to refer both to training (i.e., knowledge the staff has obtained on topics such as mental health) and communication, as several of the comments appear to refer to information received from young people rather than knowledge in the sense of formal training, and I think these should be separate as they are different concepts. Also, I’m not sure what “power” means in this context as most quotations suggest knowledge (of either type) helps support the therapeutic relationship rather than giving one party power over another. Perhaps these themes could be re-examined and clarified.

As well, there is no discussion of any strengths and limitations of this work, and the discussion is quite perfunctory (which adds to the sense of not being quite clear what the overarching findings of the paper were). Also, it might be worth mentioning the implications for practice, as this work should have clear implications for clinicians who work in these settings and I think this should be highlighted somewhere in the paper.

There is the kernel of a really interesting paper in here and the authors have done an excellent job collecting data from three different perspectives, which allows them to illuminate the same theme from various viewpoints. I would suggest revising this paper to clarify the themes up front and tighten up the structure. It is also important to address the themes that seem to “come from nowhere” – i.e., “the dance” that doesn’t seem to have been mentioned by any respondents, or “knowledge is power” where the idea of power is touched on very little. Subthemes should also be renamed to be less generic. Overall, there is a lot of value in this paper but it needs to be restructured and tightened up so it is clearer to the reader what you are trying to say.

Minor Comments:

I think the same quote has been used twice, line 341 and 406

‘you might say well that's [communication] overkill but actually this is our child, so nothing is overkill’(Carer).

This excerpt is very confusing and I’m not sure what the participant was trying to say:

"I take it like egg you know where you boil egg the water doesn't stick to the shell of the egg, then when you remove the shell, the egg from the water dries up. That’s how I see the relationship here, like both withyoung people or parents, not to take it to heart. (Staff)"

Line 577 missing apostrophe on “carers”

6. PLOS authors have the option to publish the peer review history of their article (what does this mean?). If published, this will include your full peer review and any attached files.

Reviewer #1: No

Reviewer #2: **Yes: **Sophie Wood

Reviewer #3: No

---

## [Author Response · Author response to Decision Letter 0]

26 Oct 2021

NB these are also uploaded as a Response to Reviewers document with formatting. 

Therapeutic relationships within child and adolescent mental health inpatient services: a qualitative exploration of the experiences of young people, family members and nursing staff

Thank you to the reviewers and editors for their efforts in returning this response, in what has been a very difficult year!

We welcome the thoughtful suggested and have responded to each of these below. 

We hope you will agree that the changes have improved the paper and it is now of a quality suitable for publishing in PLOS One. 

These have been updated.

We have added detail into our methods section: “The researchers liaised with clinical staff from inpatient facilities to inform them about the study. All potential participants were given a participant information sheet and the opportunity to discuss the study with the researcher, ask any questions and receive clarification. The process for obtaining informed consent was in line with HRA Guidance and a face to face, semi-structured interview was conducted by the researcher in a private room on the hospital site. Where young people were Gillick competent, we did not seek parental consent, as per our ethical approval.” 

In relation to data sharing, our consent form stipulated that:

“I give permission for direct quotes from my interview (without my name) to be shared.” 

“I understand that the findings of the study will be used to support other research in the future, and may be shared anonymously (without my name) with other researchers.” 

We therefore do not believe that we have consent to upload the data to a public repository. The data is available from the first author or the study sponsor (Pennine Care NHS Foundation Trust), Simon Kaye (Research and Innovation Manager, mailto:researchdevelopment.penninecare@nhs.net). 

The ethical statement appears only in the methods section: Participants were recruited via the UK NHS and ethical approval was granted (IRAS ID 246547).

5. Please ensure that you refer to Figures 1, 2 in your text as, if accepted, production will need this references to link the reader to the figures.

6. We note you have included a table to which you do not refer in the text of your manuscript. Please ensure that you refer to Table 1 in your text; if accepted, production will need this reference to link the reader to the Table.

These are now both referred to in the text: “An overview of participant demographics is presented in Table 1, showing that the data represents the views of a diversity of genders, ethnicities and age groups. We were also able to recruit both young people admitted informally and those on a section, alongside staff who were qualified nurses and those who were non-registered nursing assistants. Our carer group included both mothers and step-fathers. The analysis process resulted in the development of six key themes: Therapeutic relationships are the treatment, cultivating connection, knowledge is power, being human, the dance, and it’s tough for all of us in here. Figure 1 outlines the sub-theme structure and figure 2 depicts the conceptual relationship between themes. The results section is structured at the thematic level, with detail from the sub-themes woven into the analytic narrative.” 

Reviewers' comments:

Reviewer's Responses to Questions

Comments to the Author

1. Is the manuscript technically sound, and do the data support the conclusions?

Reviewer #1: Yes

Reviewer #2: Yes

Reviewer #3: Partly

2. Has the statistical analysis been performed appropriately and rigorously? 

Reviewer #1: N/A

Reviewer #2: Yes

Reviewer #3: N/A

3. Have the authors made all data underlying the findings in their manuscript fully available?

Reviewer #1: No

Reviewer #2: No

Reviewer #3: No

4. Is the manuscript presented in an intelligible fashion and written in standard English?

Reviewer #1: Yes

Reviewer #2: Yes

Reviewer #3: Yes

5. Review Comments to the Author

Reviewer #1: I very much enjoyed reading this interesting and timely paper. The approach is well-judged for the topic and the analysis shows reflexivity and sensitivity. My suggestions for improvement are as follows.

ABSTRACT

• The abstract should focus more on describing the method and results of the study (if wordcount is an issue, the first 4 sentences contain redundancy and could be condensed into 1-2 sentences). The breakdown of N=24 across participant groups should be clarified.

Detail and clarification has now been added to the abstract:

 “Child and adolescent mental health services (CAMHS), especially inpatient units, have arguably never been more in demand and yet more in need of reform. Progress in psychotherapy and more broadly in mental health care is strongly predicted by the therapeutic relationship between professional and service user. This link is particularly pertinent in child and adolescent mental health inpatient services where relationships are especially complex and difficult to develop and maintain. This article describes a qualitative exploration of the lived experienced of 24 participants (8 young people, 8 family members/carers and 8 nursing staff) within inpatient CAMHS across four sites in the UK. We interviewed participants individually and analysed the transcripts using thematic analysis within a critical realist framework. We synthesised data across groups and present six themes, encapsulating the intricacies and impact of therapeutic relationships; their development and maintenance. We hope these findings can be used to improve quality of care by providing a blueprint for policy, training, systemic structures and staff support.” 

INTRODUCTION

• The Introduction seems to be specific to the UK context (e.g. 2nd sentence references NHS commitment to improved access to community services) – this should be stated at the very start.

This has been clarified: “Internationally, inpatient care faces huge challenges alongside growing demand [5]. Particularly in the UK, where this paper is focused, there is a drive for quality improvement.”

• Line 65: “manage the ‘push and pull’ of working with adolescents” – meaning unclear

This has been clarified: “When supporting young people, there is a need to navigate issues of power and control [23] and manage the complexities, emotional and relational experiences or ‘push and pull’ of working with adolescents who experience disrupted ways of relating to people due to their early relationships [24].”

• Page 3 states “there has been no in-depth study reporting on the alliance in this context from the perspectives of all key stakeholders.” This is somewhat unclear to me. Do the authors mean that no previous research has taken a qualitative approach to explore the therapeutic alliance? Or simply that no studies have involved multiple participant groups in the same analysis? If the former, this should be more clearly stated. If the latter, the authors should explain why combining multiple groups into the same analysis is important or valuable, above and beyond a collection of individual studies that each focus on one stakeholder group. Additionally, if previous studies on the therapeutic alliance have explored the unique first-person perspectives of young people/staff/carers, the key findings of such research should be outlined in the Introduction.

This has been clarified: “there has been no in-depth qualitative study of the alliance in this context. The current study therefore aimed to comprehensively explore the experience of therapeutic relationships between young people, their carers and nursing staff admitted to or working on adolescent inpatient mental health wards. This is the first study to qualitatively investigate the alliance in this setting from the multiple perspectives involved in therapeutic relationships here, enabling the synthesis of a rich dataset that permits a more in-depth understanding and thus provide material to aid the development of interventions or contexts to support future service improvements.”

• The Introduction should do more to articulate the rationale for a qualitative approach to this topic, and for a critical realist epistemology specifically.

Detailed added here: “The qualitative approach enables the synthesis of a rich dataset that permits a more in-depth understanding and thus might provide material to aid the development of interventions or contexts to support future service improvements.”

And here: 

“The study proceeded in line with a critical realist perspective, which allows the influence of the specific context of inpatient units to be recognised.”

With more detail in the methods section:

“The study proceeded in line with critical realism and contextualism perspectives. A critical realist epistemological stance was taken as this is most consistent with the aims and context of the study. This stance assumes that psychological phenomena do have some external basis in reality outside of any single individual’s interpretation, but these phenomena are blurred and inherently bounded by culture and context; arguably especially important in settings such as inpatient wards and thus requiring careful consideration [46]. This framework allows broad inferences to be drawn whilst recognising the particular context of the participants [47].”

METHOD

• Were the participants related to each other? E.g. were the young people and carers from the same family, or did the staff provide care to the specific young people interviewed?

In terms of family systems, participants were not necessarily related to each other, no. We neither encouraged nor prevented this, which is an approach that was approved by our ethics committee review. We are aware that a total of four participants (two carers and two young people) were related to one another. In terms of staffing relationships, participants (staff, young people and carers) were recruited across four sites and so yes, some staff will have cared for the young people interviewed. The nature of inpatient care is that all staff members will have contact/ relationships of varying degrees with all young people and family members. 

• The coding process should be more explicitly reported. The statement that analysis was performed “utilising paper, Word and Excel” raises questions about how systematic/consistent the process was. The coding frame should be described e.g. with number, hierarchical structure, examples of codes. Were all participant groups coded using the same coding frame? Was coding conducted by line, paragraph or purposively selected sections of text? How were codes developed into themes? What steps were taken to ensure the credibility of the analysis?

Detail was reduced to streamline this section. We have now added this in on the reviewer’s request:

“The interviews were recorded, transcribed, coded and analysed using thematic analysis [43-45] utilising paper, Word and Excel. Considering the limited theory in the field, the development of themes was inductive in nature, without reference to existing theories and semantic, manifest meaning was prioritised. Although it is recognised the boundary with more latent meaning is not always clear. The first author led the analysis, alongside the second, with review from the third. The analysis proceeded in the following manner: i) Authors 1 and 2 familiarised themselves with one transcript (randomly selected); ii) Authors 1 and 2 both independently coded this transcript line by line then met to review and discuss; iii) On the basis of this discussion, it was decided that only data directly relevant to the research question be coded; iv) Author 2 then independently coded a second randomly selected transcript and reviewed this with author 1, from which initial codes were identified; v) Author 2 then coded the remaining data, utilising, adding to and adapting the initial codes; vi) Authors 1 and 2 met regularly to review this coding and the data extracted; vii) Using printed paper versions of the codes, authors 1 and 2 reviewed and grouped related codes together to explore linkages and overlap in order to produce themes and subthemes; viii) Authors 1 and 2 then reviewed these in relation to the original codes, making any necessary amendments, ix) Author 2 then loosely filtered the quotes for inclusion within the subthemes, omitting data that was not relevant to the research question or not related to nursing staff; x) Author 1 filtered this further, liberally selecting the most relevant quotes for each subtheme; xi) Author 2 reviewed these selected quotes for homogeneity with the subtheme and theme definitions.” 

• Table 1 – please also include age ranges

We weren’t sure whether you meant mathematical range or both the maximum and minimum values. We have now included the latter but can amend further:

Participant demographic data is detailed below in Table 1:

Carer group (n=8)

Gender and role Female mother (6) Male step-father (2)

Mean age 48 (SD 7.59, range 36-56)

Ethnicity White British (7) British Pakistani (1)

Staff member group (n=8)

Gender Female (6) Male (2)

Mean age 39 (SD 9.28, range 26-55)

Ethnicity White British (6) Black African (1) Mixed Race (1)

Job role Nurse (3) Nursing Assistant (5)

Mean years working in inpatient setting 7 (SD 6.36)

Young Person (YP) group (n=8)

Gender Female (5) Male (3)

Mean age 15.62 (SD: 1.31, range 13-17)

Ethnicity White British (7) British Pakistani (1)

Mental Health Act (1983) status Detained under section (2) Informal (6)

RESULTS

• The Results show excellent analytic insight and the figures are very helpful in navigating the complex content. However, this section is very long and I think the authors should consider streamlining it. It is demanding a lot of readers to digest this quantity of complex material. The length means that important findings are getting ‘lost in the crowd’: after finishing the paper, I was not left with any solid sense of what its specific core contributions were.

A previous review of this paper by a qualitative-specific journal advised us to add in detail to the results section to ensure it incorporated both the presentation of the main themes and the analysis of their layers and complexities. We have restructured and added to the discussion section as suggested below to ensure that the headline findings are clear for the reader. 

• Line 587-590 – sentence long and difficult to follow

This has been restructured into separate sentences and the wording adapted to aid clarity:

“This taking on of responsible can exacerbate the emotional burden that staff experience. In addition there’s a fear that it could render young people overly reliant on others, at risk of becoming deskilled or disempowered and limiting the normative adolescent development towards independence. Staff members have to navigate these complexities, balancing the need for care and independence, for safety and empowerment.”

DISCUSSION

• The Discussion needs further development. It should include (a) a concise summary of the key take-home points of the research, (b) more explicit articulation of what the findings add to existing literature i.e. how they reinforce/contradict/expand on previous research, (c) more consideration of the real-world implications of the research (e.g. the importance of providing staffing resources that acknowledge the therapeutic importance of casual ‘down-time’ and staff emotional support), (d) critical reflection on the strengths and limitations of the study, with suggestions for future research.

Thank you for highlighting this and making such thoughtful suggestions of possible additions, we have added to, restructured and improved the discussion throughout. 

• I was somewhat surprised to see no mention of continuity of care in the Intro, Results or Discussion. If the therapeutic relationship is so important, presumably it is important that young people are dealing with the same staff on an ongoing basis. Of course this is less an issue in inpatient than community services. But there is mention of agency staff in the Results – would these people just enter the unit on a very temporary basis and how does that impact on relationships? Might the Discussion consider any implications for continuity of care/facilitating ongoing contact with staff after young people leave the inpatient unit?

We have added to this in the discussion: “Consideration should also be given to aspects of the workforce that might impact on this being successful, such as the role of agency or bank staff, where continuity of care and relationships might be impeded.”

“Admissions, transitions and endings might be points of particular vulnerability and so paying attention to relationships at these times will be key.”

Reviewer #2: Summary

This paper presents a qualitative exploration of the role of therapeutic relationships in Child and Adolescent Mental health Inpatient Services from the perspective of service users, their family/carers and the professionals who provide these services. The contribution to the field is welcome and timely given the increasing demand for these services, which is anticipated to be exacerbated by the Covid-19 restrictions. By better understanding the importance of the role of therapeutic relationships, service delivery can be improved which could result in better outcomes for the young people involved, thereby improving their wellbeing and future opportunities in adulthood. The paper therefore has considerable practice and policy implications. The insight and time provided by these young people, their parents/carers and nursing staff is invaluable, and it would be unethical not to publish such findings. There is also a strong argument for the paper backed up by relevant literature, and the themes identified are enlightening and supported by the data presented. However, the manuscript is lacking some basic elements which would expected from an academic journal article. Therefore, I suggest a few revisions and encourage the authors to re-submit.

Thank you for your strong endorsement of the value and pertinence of the study. 

Major issues

Findings: You need to start the findings section outlining what your 6 key themes are and the order in which you will present them (and reference fig. 1). Then you need a small section on participant characteristics with reference to Table 1.

This has now been added: 

“An overview of participant demographics is presented in Table 1, showing that the data represents the views of a diversity of genders, ethnicities and age groups. We were also able to recruit both young people admitted informally and those on a section, alongside staff who were qualified nurses and those who were non-registered nursing assistants. Our carer group included both mothers and step-fathers. The analysis process resulted in the development of six key themes: Therapeutic relationships are the treatment, cultivating connection, knowledge is power, being human, the dance, and it’s tough for all of us in here. Figure 1 outlines the sub-theme structure and figure 2 depicts the conceptual relationship between themes. The results section is structured at the thematic level, with detail from the sub-themes woven into the analytic narrative.” 

Discussion: Missing implications for future research and you need to be more specific about the implications for policy and practice. You could provide recommendations.

Discussion: Missing study strengths and limitations section.

We have significantly restructured and added to the discussion section, thank you for highlighting this and making suggestions. 

Minor issues

Line 102 under “Sample”: It reads as though you only interviewed 8 YP, carers and nurses in total. Make it clear that you interviewed 8 of each and include the total sample size (24).

This has been amended: “We recruited eight young people (YP), eight carers and eight nursing staff. 

Line 127: How did you meaningfully engage the young people in developing the topic guide?

This detail has been added: “Topic guides were developed from published literature and in consultation with the patient, carer and staff involvement group allied to the project, who reviewed drafts, suggested wording changes and additional areas of exploration and prompts.”

Line 142- 145: Your reflection on the first author’s relationship with participants is good, but could you provide an example regarding your reference to inpatient wards, relevant to the study, and explain how you carefully considered this?

We have added to this in the methods section here: “Participants were made aware that the interviewer was a clinical psychologist and researcher working in an adolescent inpatient facility; the potential for this to influence the discourse (in terms of a staff member asking about relationships with staff members) was acknowledged with participants and they were reminded of confidentiality and encouraged to share as openly as they could.”

Line 172: Explicitly state this is the first theme in the subtitle e.g. “Theme 1: Therapeutic Relationships are the Treatment”

Thank you for suggesting this. All theme labels have now been numbered. 

Quotes: Number the participants so the reader can see if it is the same or a different participant with a certain view (e.g. YP1, Staff1 etc.).

Participant identification numbers and pseudonyms are not advised, e.g. by publications such as Qualitative Health Research as these can threaten anonymity. The quotes come from the full range of participants. 

Figures and tables are not referenced in the text- please reference and discuss them.

These are now referenced:

“An overview of participant demographics is presented in Table 1. Figure 1 outlines the sub-theme structure and figure 2 depicts the conceptual relationship between themes. The results section is structured at the thematic level, with detail from the sub-themes woven into the analytic narrative.” 

Table 1 is unclear and needs formatting, e.g. the headings for carer group, staff carer group, young person group are not aligned. You use (n=x) for some totals but (x) for others. I think it would be clearer if you had the participant groups as columns at the top and characteristics as rows down the side. If some groups do not share a characteristic, leave it blank. It is very difficult to read.

Thank you for the suggestion, this has now been restructured and reformatted:

 Carer (n=8) Staff member (n=8) Young person (YP; n=8)

Gender Female (n=6)

Male (n=2) Female (n=6) 

Male (n=2) Female (n=5) 

Male (n=3)

Mean age 48 

(SD 7.59, range 36-56) 39 

(SD 9.28, range 26-55) 15.62 

(SD: 1.31, range 13-17)

Ethnicity 

(self-described) White British (n=7) British Pakistani (n=1) White British (n=6) Black African (n=1) Mixed Race (n=1) White British (n=7) British Pakistani (n=1)

Role Mother (n=6)

Step-father (n=2) Nurse (n=3)

Nursing Assistant (n=5) -

Mean years working in inpatient setting - 7 (SD 6.36) -

Mental Health Act status - - On section (n=2) 

Informal (n=6)

Reviewer #3: This is a generally well-written paper which covers several topics around child and adolescent mental health inpatient services. One of the major strengths is the inclusion of interviews with staff members, young people using the service, and carers or family members. This has allowed for a more well-rounded understanding of the topic and the authors have brought in multiple perspectives to illustrate each theme.

Thank you – we agree that this is a key strength of the paper. 

However, the structure of the paper lets it down somewhat. There is very little sense of coherence or how the themes inter-relate, and I think is partly because there are a large number of themes and subthemes, some of which don’t necessarily seem to come from the data. The themes are not even listed in the abstract, which makes it difficult to get a sense of the main argument of the paper. Each theme has a lot of ideas contained in it and it is not always clear how they are related – and some quotations used don’t seem to illustrate the point that is being made.

We have added statements to both the abstract and results to aid clarity. The figures are also available to depict the sense of the interrelatedness of the themes and subthemes, and these are now referred to in the text. 

“We interviewed participants individually and analysed the transcripts using thematic analysis within a critical realist framework. We synthesised data across groups and present six themes, encapsulating the intricacies and impact of therapeutic relationships; their development and maintenance: Therapeutic relationships are the treatment, cultivating connection, knowledge is power, being human, the dance, and it’s tough for all of us in here. We hope these findings can be used to improve quality of care by providing a blueprint for policy, training, systemic structures and staff support.” 

“An overview of participant demographics is presented in Table 1, showing that the data represents the views of a diversity of genders, ethnicities and age groups. We were also able to recruit both young people admitted informally and those on a section, alongside staff who were qualified nurses and those who were non-registered nursing assistants. Our carer group included both mothers and step-fathers. The analysis process resulted in the development of six key themes: Therapeutic relationships are the treatment, cultivating connection, knowledge is power, being human, the dance, and it’s tough for all of us in here. Figure 1 outlines the sub-theme structure and figure 2 depicts the conceptual relationship between themes. The results section is structured at the thematic level, with detail from the sub-themes woven into the analytic narrative. “ 

The themes all have sub-themes, which is helpful in understanding the data – but they are very generic/vague. For example, a sub-theme of the last theme is “what makes it hard” – someone just looking at the theme structure would have no idea what that actually meant. Instead, you could name the subthemes more specifically (e.g, “racism”, “boredom”, “emotional toll on staff”) so it is clear what your themes relate to.

The subthemes are themselves made up of a number of codes so they inherently relates to a number of different ideas/ experiences, as per the process of thematic analysis. We have relabelled some of them as per the new figure:

A few of the themes did not seem to have their genesis in the data –for example, the metaphor of “the dance” did not seem to be mentioned by any participant and appeared to be superimposed by the authors, but this wasn’t explicitly acknowledged. (Or if this came from elsewhere – for example, if there was a theoretical basis for using this term - it would be helpful to acknowledge that as well.) Also, it just seems a strange metaphor to use if it didn’t come from the data – “Young people, carers and staff have to occupy different positions and roles over the course of the admission; playing different parts” – do people necessarily play different roles and parts during the course of a dance? Also the topics explored in these theme don’t seem to have much to do with the name “dance” (e.g., the topic of responsibility etc). You do refer to the idea of a “triangle” which DID come directly from the data – perhaps this would be a better theme?

As the reviewer has inevitably not had access to the full data, nor opportunity to engage in the analytical process, we accept that they might have a different interpretation of the ideas presented, or struggle to trust that they have emerged from the data. We are sorry that some of the themes didn’t seem clear; they have been well received by our participation groups, readers of the pre-print and viewers of our animated summary. Metaphors can often better represent multi-layered concepts better than simple descriptions. The data that informed this theme included reference to movement, alongside different ‘protagonists’ with differing roles. In dance, frequently one person will ‘take the lead’; therefore the metaphor also alludes to responsibility. We think that this is well reflected by the theme name, which seeks to capture the richness of the data as a whole rather than individual quotes or codes. We agree that the triangle depicts well the three key players in this concept but it does not capture the movement over space and time. As ever with qualitative analysis, finding one specific quote that captures a whole theme perfectly is unlikely as the theme is made up of subthemes and codes and is in a sense more than the sum of its parts. 

Similarly, the theme “Knowledge is Power” is a bit of a strange summing up of the qualitative quotes in that section as very few of them actually speak of power, and “knowledge” seems to refer both to training (i.e., knowledge the staff has obtained on topics such as mental health) and communication, as several of the comments appear to refer to information received from young people rather than knowledge in the sense of formal training, and I think these should be separate as they are different concepts. Also, I’m not sure what “power” means in this context as most quotations suggest knowledge (of either type) helps support the therapeutic relationship rather than giving one party power over another. Perhaps these themes could be re-examined and clarified.

This theme speaks to the power that knowledge has in the development and maintenance of therapeutic relationships. The development of knowledge and understanding (which includes both professional expertise and person-centred knowledge) and the sharing of information and understanding between parties were both highlighted as crucial. Participants frequently spoke of needing both professional and personal knowledge and that is why these are not separated out. We have altered the sub theme title to reflect this: 

As well, there is no discussion of any strengths and limitations of this work, and the discussion is quite perfunctory (which adds to the sense of not being quite clear what the overarching findings of the paper were). Also, it might be worth mentioning the implications for practice, as this work should have clear implications for clinicians who work in these settings and I think this should be highlighted somewhere in the paper.

This has now been added and the discussion in general has been substantially reworked. 

“The convergence of themes (alongside the diversity of individual views) across young people, carers and nursing staff reflects the triumvirate nature of the experience [28] and the value in synthesising different perspectives along a common thread. Future research could address the limitations of the current study by synthesising the experiences of additional professional groups alongside nursing staff and ensuring better representation of diversity for all groups in terms of gender, ethnicity and marginalised identities.”

There is the kernel of a really interesting paper in here and the authors have done an excellent job collecting data from three different perspectives, which allows them to illuminate the same theme from various viewpoints. I would suggest revising this paper to clarify the themes up front and tighten up the structure. It is also important to address the themes that seem to “come from nowhere” – i.e., “the dance” that doesn’t seem to have been mentioned by any respondents, or “knowledge is power” where the idea of power is touched on very little. Subthemes should also be renamed to be less generic. Overall, there is a lot of value in this paper but it needs to be restructured and tightened up so it is clearer to the reader what you are trying to say.

Minor Comments: I think the same quote has been used twice, line 341 and 406

‘you might say well that's [communication] overkill but actually this is our child, so nothing is overkill’(Carer).

You’re right, this is intentional as it is included as the illustrative quote in the theme title and (as a longer version) forms part of the in depth analysis of that theme. 

This excerpt is very confusing and I’m not sure what the participant was trying to say:

"I take it like egg you know where you boil egg the water doesn't stick to the shell of the egg, then when you remove the shell, the egg from the water dries up. That’s how I see the relationship here, like both with young people or parents, not to take it to heart. (Staff)"

This was a beautiful quote, which was offered by someone whose first language was not English. I asked them to clarify following the quote, which they did and explained that when boiling an egg in its shell if you remove it from the water, the surface dries immediately. So it illustrates that staff need to let things wash over them quickly – be impervious to emotional distress. Another version might be ‘like water off a duck’s back’. 

Line 577 missing apostrophe on “carers”

Thank you- this has been added.

---

## [Editor Report · Decision Letter 1]

1 Dec 2021

PONE-D-20-31788R1Therapeutic relationships within child and adolescent mental health inpatient services: a qualitative exploration of the experiences of young people, family members and nursing staffPLOS ONE

Dear Dr. Hartley,

Thank you for submitting your manuscript to PLOS ONE. I have been invited to serve as Guest Academic Editor for your manuscript. For the sake of transparency, please note that I participated in the original evaluation of your manuscript (Reviewer 1).  Thank you for your thorough revision in light of reviewers' comments. I believe your manuscript is nearly ready for publication; however as PLOS ONE does not provide copy-editing services, the manuscript requires a careful proof-read before formal acceptance. Some (relatively minor) issues that require attention are listed below. We invite you to submit a revised version of the manuscript that addresses the points raised.

(Please note line numbers reflect the track-changed manuscript)

Line 29-32 – the punctuation in this sentence is confusing. It would be helpful to demarcate the theme titles with capitalisation, quote marks or italics (similarly in line 214-216)Line 49 – should not start a sentence with ‘But’Line 121 – state the official name of the ethics committee that approved the researchLine 180-183 – inconsistent capitalisation of ‘author’Line 187 – “omitting data that was not relevant to the research question or not related to nursing staff” – this is not clear as you have not explicitly stated a specific research question, and there is no indication of what is meant by ‘not related to nursing staff’ given that not all participants were nursesLine 216 – capitalise Fig 2Line 643, 735, 859 – avoid contractions (unless a direct quote)Lines 653-654, 724 – inappropriate apostrophesPlease ensure consistency throughout the Results section regarding whether quotes on new lines are preceded by a colon or semi-colon.Minor typos/errors in lines 429, 552, 562, 658, 661, 673, 764, 814, 859

Additionally, certain reviewer comments have not been directly addressed within the revised manuscript. Specifically:

R1’s query about the relationships between participants should be clarified within the Sample sectionR2’s request for quotes to be accompanied by participant ID numbers is appropriate and consistent with COREQ guidelines and PLOS ONE norms for qualitative articlesR3’s request for clarification of the quote about the ‘egg’ is appropriate – the meaning of this quote is not accessible to a reader without the type of contextualisation provided in the response to reviewersPlease submit your revised manuscript by Jan 15 2022 11:59PM. If you will need more time than this to complete your revisions, please reply to this message or contact the journal office at plosone@plos.org. Please include the following items when submitting your revised manuscript:A rebuttal letter that responds to each point raised by the academic editor and reviewer(s). You should upload this letter as a separate file labeled 'Response to Reviewers'.A marked-up copy of your manuscript that highlights changes made to the original version. You should upload this as a separate file labeled 'Revised Manuscript with Track Changes'.An unmarked version of your revised paper without tracked changes. You should upload this as a separate file labeled 'Manuscript'.If applicable, we recommend that you deposit your laboratory protocols in protocols.io to enhance the reproducibility of your results. Protocols.io assigns your protocol its own identifier (DOI) so that it can be cited independently in the future. For instructions see: https://journals.plos.org/plosone/s/submission-guidelines#loc-laboratory-protocols. Additionally, PLOS ONE offers an option for publishing peer-reviewed Lab Protocol articles, which describe protocols hosted on protocols.io. Read more information on sharing protocols at https://plos.org/protocols?utm_medium=editorial-email&utm_source=authorletters&utm_campaign=protocols.

We look forward to receiving your revised manuscript.

Kind regards,

Cliodhna O'Connor

Academic Editor

PLOS ONE
---

## [Author Response · Author response to Decision Letter 1]

13 Dec 2021

Full details given in response to reviewers document as requested.

---

## [Editor Report · Decision Letter 2]

17 Dec 2021

Therapeutic relationships within child and adolescent mental health inpatient services: a qualitative exploration of the experiences of young people, family members and nursing staff

PONE-D-20-31788R2

Dear Dr. Hartley,

We’re pleased to inform you that your manuscript has been judged scientifically suitable for publication and will be formally accepted for publication once it meets all outstanding technical requirements.

Kind regards,

Cliodhna O'Connor

Guest Editor

PLOS ONE
---

## [Editor Report · Acceptance letter]

6 Jan 2022

PONE-D-20-31788R2 

Therapeutic relationships within child and adolescent mental health inpatient services: a qualitative exploration of the experiences of young people, family members and nursing staff 

Dear Dr. Hartley:

I'm pleased to inform you that your manuscript has been deemed suitable for publication in PLOS ONE. Congratulations! Your manuscript is now with our production department. 

Kind regards, 

on behalf of

Dr. Cliodhna O'Connor 

Guest Editor

PLOS ONE